# Efficient optimization with higher-order Ising machines

Connor Bybee [1] ✉, Denis Kleyko [1,2], Dmitri E. Nikonov [3],
Amir Khosrowshahi[1,3], Bruno A. Olshausen[1] & Friedrich T. Sommer [1,4] ✉

A prominent approach to solving combinatorial optimization problems on parallel hardware is Ising machines, i.e., hardware implementations of networks of interacting binary spin variables. Most Ising machines leverage second-order interactions although important classes of optimization problems, such as satisfiability problems, map more seamlessly to Ising networks with higher-order interactions. Here, we demonstrate that higher-order Ising machines can solve satisfiability problems more resource-efficiently in terms of the number of spin variables and their connections when compared to traditional second-order Ising machines. Further, our results show on a benchmark dataset of Boolean $k$-satisfiability problems that higher-order Ising machines implemented with coupled oscillators rapidly find solutions that are better than second-order Ising machines, thus, improving the current state-of-the-art for Ising machines.

An Ising machine is a type of parallel computer utilizing energy relaxation in a network of interacting binary variables. Ising machines have been proposed as efficient methods for finding optimal or near-optimal solutions to hard combinatorial optimization problems[1–6]. For a given combinatorial optimization problem, the network interactions are shaped so that the energy minima correspond to the problem solutions. For mapping a given combinatorial optimization problem to a network, a common strategy is to formulate the objective as the energy function of an Ising model, an abstract network of coupled bipolar variables originally proposed to model ferromagnetic material. The Ising model can then be implemented on hardware, referred to as an Ising machine. Ising machines implemented on quantum computers promise optimal solutions[3,7–9]. However, due to the challenges of constructing them, Ising machines based on classical physics are reemerging and new technologies are being developed. There is a large variety of possibilities for implementing classical Ising machines, including coupled electrical oscillators[6,10–12], optical parametric oscillators[13], stochastic circuits (probabilistic bits)[14], and neuromorphic hardware[1,15,16]. Here, we focus on classical Ising machines for approximately solving combinatorial optimization problems at scale and extremely fast.

Casting a combinatorial optimization problem as an Ising model usually takes two or three steps. The first step is to express the combinatorial optimization problem objective as a polynomial in the binary variables. The second step is mapping the polynomial to the energy function of an Ising model. For many combinatorial optimization problems, step one results in a higher-order polynomial[2,17–21], i.e., a polynomial with terms that contain products of more than two binary variables. However, most Ising machines utilize second-order polynomial interactions between variables. In this case, a third step, called quadratization[2,17,18,20,22–25], is applied for reducing higher-order terms in the polynomial to second-order. The resulting second-order polynomial represents the energy function of a classical Ising model, i.e., a second-order network in which each interaction just couples a pair of variables[26]. Quadratization increases the network size by adding auxiliary variables and it requires increased precision and range of the second-order interaction coefficients compared to higher-order interactions[18,19].

Higher-order Ising models—models that include polynomial interactions of a degree greater than two—have received little attention because the possible number of interactions grows exponentially with the interaction order. Thus, the training and implementation of

[1]Redwood Center for Theoretical Neuroscience, University of California, Berkeley, CA, USA. [2]Intelligent Systems Lab, Research Institutes of Sweden, Kista, Sweden. [3]Components Research, Intel, Hillsboro, OR, USA. [4]Neuromorphic Computing Lab, Intel, Santa Clara, CA, USA.
✉ e-mail: bybee@berkeley.edu; fsommer@berkeley.edu

higher-order Ising models seemed intractable and impractical[27]. Here, we propose to skip the step of quadratization and instead use higher-order Ising models that directly implement the higher-order polynomials describing the combinatorial optimization problems. Although this proposal seems daunting at first glance, we show that for important classes of combinatorial optimization problems, the corresponding higher-order Ising machines require fewer variables and connections than the second-order Ising machines resulting from the quadratization approach.

Among the proposed Ising machines, coupled electrical oscillators are promising for combinatorial optimization problems[28] in terms of solution quality[29], and the ability to leverage existing technologies such as complementary metal-oxide-semiconductor (CMOS) ring oscillators[30,31]. Further, the multiplication and routing of electrical signals that are required to implement a $k$-th-order interaction for an arbitrary order $k$ can be realized with existing technologies commonly used in devices such as phase detectors and mixers[32–34], offering advantages over other physical systems[35]. To build an oscillator Ising machine, the continuous phases of oscillator variables have to be biased towards two anti-symmetric states, for example, by sub-harmonic injection locking[29,36,37]. To demonstrate a concrete higher-order Ising machine, we investigate a network of coupled Hopf oscillators with sub-harmonic injection locking, referred to as a higher-order oscillator Ising machine. Results from our simulations show that the higher-order oscillator Ising machine not only uses fewer network resources compared to the second-order oscillator Ising machine but, importantly, achieves better solutions. All told, our results suggest that, against common beliefs, optimization with higher-order Ising machines can outperform traditional Ising model approaches.

## Results

### Mapping constraint satisfaction problems to Ising models

A broad class of combinatorial optimization problems are constraint satisfaction problems, including invertible logic circuits, Boolean satisfiability (SAT) problems, and Boolean maximum satisfiability (MaxSAT) problems. SAT solvers have many direct applications in areas, such as artificial intelligence[38], electronic design automation[39], cryptography[40], and many more. Many Boolean constraint satisfaction problems naturally map to higher-order polynomials[2,17]. The most common approach for solving constraint satisfaction problems with Ising machines has been first to apply quadratization for translating problems to second-order polynomials, and then use second-order Ising machines to solve them efficiently[2,18,19,22,24,26]. However, optimization can also be performed in higher-order Ising machines without quadratization[21,41,42]. Here, we aim to construct higher-order Ising machines for Boolean constraint satisfaction problems which are simple, yet, scale to large problems and quickly find near-optimal solutions.

In Boolean constraint satisfaction problems, the Boolean variables must take a state which satisfies a set of pre-defined constraints. For the $h$-th constraint containing $k$ variables, the state space, $\mathbf{S}_h = \{-1, 1\}^k$, can be partitioned into two sets. Let $\mathbf{C}_h = \{\mathbf{c} \in \mathbf{S}_h : \mathbf{c} = \text{satisfied state}\}$ be the set of valid states, i.e., that satisfy the constraint, and $\bar{\mathbf{C}}_h = \mathbf{S}_h \setminus \mathbf{C}_h = \{\mathbf{c} \in \mathbf{S}_h : \mathbf{c} = \text{unsatisfied state}\}$ be the set of invalid states which do not satisfy the constraint. Any logic function can be expressed by a constraint for which the set $\mathbf{C}_h$ represents the truth table of the function. An objective or energy function of the $h$-th constraint, $E_h$, can be written as the characteristic function of its set of invalid states[2]:

$$E_h(\mathbf{s}) = \sum_{\mathbf{c} \in \mathbf{C}_h} \prod_{i=1}^{k} (1 + c_i s_i)/2 \qquad (1)$$

or, equivalently (Methods, Equivalence of higher-order Ising energy formulations), as one minus the characteristic function of its set of valid states:

$$E_h(\mathbf{s}) = 1 - \sum_{\mathbf{c} \in \mathbf{C}_h} \prod_{i=1}^{k} (1 + c_i s_i)/2. \qquad (2)$$

Thus, the sizes of the sets of valid and invalid states may determine which of the two equations is preferable. Let $N_{\mathbf{C}_h} = |\mathbf{C}_h|$ and $N_{\bar{\mathbf{C}}_h} = |\bar{\mathbf{C}}_h|$ denote the size of the set $\mathbf{C}_h$ and $\bar{\mathbf{C}}_h$, respectively. Then, Eqs. (1) and (2) contain a sum with $N_{\bar{\mathbf{C}}_h}$ and $N_{\mathbf{C}_h}$ terms, respectively. Note that both energies contain higher-order interactions of the order of the size of the constraint.

The total energy for a constraint satisfaction problem is the weighted sum of the individual constraints, Eq. (3):

$$E(\mathbf{s}) = \sum_{h \in \boldsymbol{\Gamma}} w_h E_h(\mathbf{s}). \qquad (3)$$

Equation (3) generalizes our method to weighted MaxSAT problems, which have many applications[43]. In MaxSAT, each constraint is assigned a weight, $w_h$, representing the relative importance of satisfying the $h$-th constraint. Here, $\boldsymbol{\Gamma}$ is the set of indices for the problem constraints, $E_h$ is the energy function for the $h$-th constraint formulated according to either Eq. (1) or (2).

Equation (1) or (2) are higher-order interactions represented as factored polynomials. Equation (3) can be expanded to coincide with the common formulation of a higher-order Ising model

$$E(\mathbf{s}) = - \left( J^{(0)} + \sum_{i_1} J^{(1)}_{i_1} s_{i_1} + \sum_{i_1 < i_2} J^{(2)}_{i_1 i_2} s_{i_1} s_{i_2} + \ldots + \sum_{i_1 < \ldots < i_k} J^{(k)}_{i_1 \ldots i_k} s_{i_1} \ldots s_{i_k} + \ldots + \sum_{i_1 < \ldots < i_n} J^{(n)}_{i_1 \ldots i_n} s_{i_1} \ldots s_{i_n} \right). \qquad (4)$$

Here the real-valued variable $J^{(k)}$ represents the $k$-th order interaction between $k$ spin variables and $n$ is the total number of spin variables in the Ising model. The first three terms with 0-th to 2-nd order interactions of (4) form the energy function of the traditional Ising model. Only small subsets of all possible interactions will be present for a particular optimization problem. Either the factored or expanded parameterization may be preferred depending on the problem and which form results in the fewest number of terms in the energy. In general, the expanded energy may contain $2^k - 1$ terms or parameters. However, for many practical problems, each clause contains only a few literals, hence, $k$ is small. The factored representations require $N_{\mathbf{C}_h} k$ and $N_{\bar{\mathbf{C}}_h} k$ parameters for Eqs. (2) and (1), respectively. Thus, when $k$ is large or the expanded form does not simplify to a few terms, the factored representation is preferable.

The derivation of Ising models is first explained for two small examples of combinatorial optimization problems, the exclusive OR (XOR) invertible logic gate, and a small SAT problem. The XOR problem can be depicted by the XOR gate symbol (Fig. 1a), and its state table (Fig. 1b). The expanded and simplified energy polynomial of XOR contains only one interaction (Fig. 1c), resulting in a very simple hypergraph of the corresponding third-order Ising network (Fig. 1d). The quadratization of the third-order XOR polynomial produces a second-order Ising network with one additional auxiliary variable, six second-order interactions, and four biases (Fig. 1e). The additional network resources required after quadratization may be negligible for small problems but significantly change the scaling behavior of required resources for larger problems (Fig. 2).

Any SAT problem can be written as the product (conjunction or AND) of clauses (constraints) where each clause is the Boolean sum (OR) of literals. A literal is a variable or its negation. This form is known as conjunctive normal form (CNF). For a particular 3 clause SAT problem, the CNF (Fig. 1f) corresponds to a logic gate circuit (Fig. 1g), and a factored higher-order energy polynomial (Fig. 1h). The factored energy

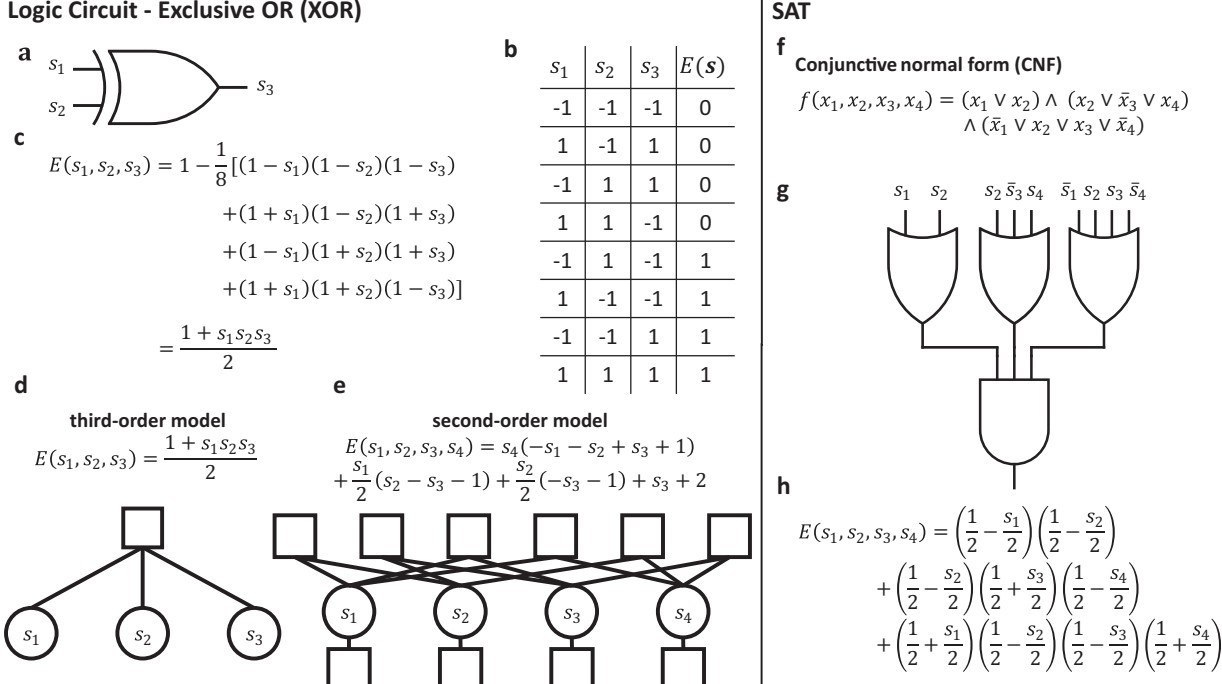

**Fig. 1 | Mapping optimization problems to Ising models.** Two example problems. Left: XOR circuit. **a** Circuit schematic for the XOR. The XOR gate has two inputs, $s_1$ and $s_2$, and one output, $s_3$. **b** The state table has eight lines. Four lines are input configurations for valid/true output, the other four are input configurations for invalid/false output. **c** The higher-order energy function for the XOR in both the factored and simplified form. **d** Energy and corresponding hypergraph of third-order XOR Ising network, variables nodes, depicted as circles, connected by one interaction, depicted as a square. **e** Energy and corresponding graph of second-order XOR Ising network, resulting from quadratization. The graph contains four variable nodes (one auxiliary variable), six second-order interactions, and four first-order interactions (biases). Right: SAT problem. **f** SAT problem in CNF. The SAT function is written with binary variables, $x_i \in \{0, 1\}$, where $\bar{x}_i$ denotes the variable negation. **g** The SAT problem in CNF has an equivalent circuit representation consisting of $k$-input OR gates which output to one AND gate. **h** The energy can be succinctly formulated with one term per clause using Eq. (1).

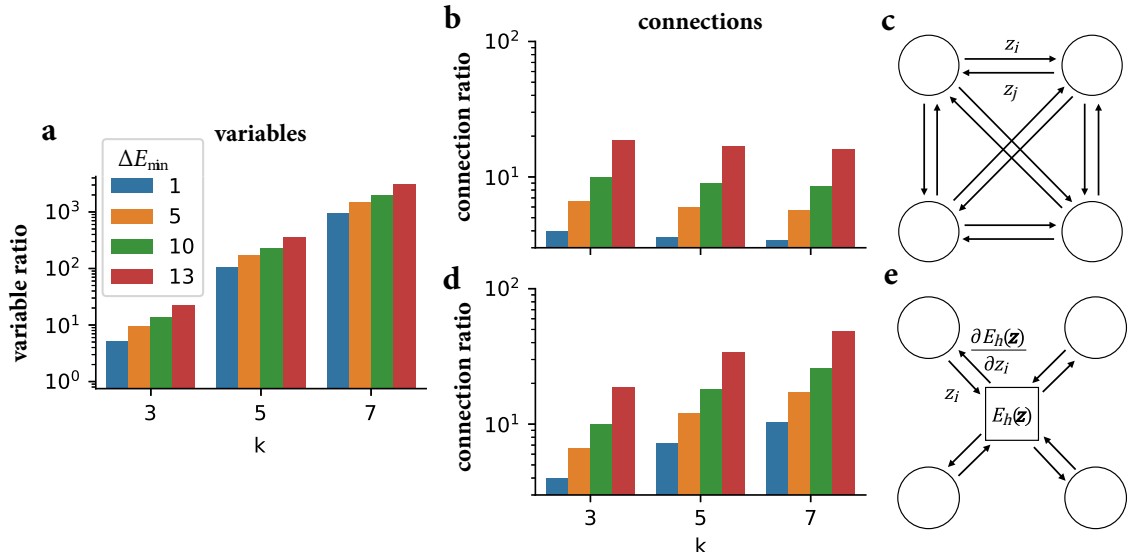

**Fig. 2 | Comparing second-order to third-order model parameters on benchmark $k$SAT problems. a, b, d** The ratio between the number of second-order parameters to higher-order parameters are plotted as a function of the number of variables per constraint in the $k$SAT problem and different values of $\Delta E_{min}$. Colors represent reductions with a different minimum energy gap, $\Delta E_{min}$. The bars are grouped by $k$, the number of variables per clause. **a** The ratio of the number of variables required for second-order networks compared to higher-order networks. **c** A higher-order interaction implemented with all-to-all connectivity. **e** A higher-order interaction is implemented with a computational node for each constraint. **b, d** The ratio of the number of connections required for second-order networks compared to higher-order networks implemented with all-to-all connectivity (**c**) and intermediate computational nodes (**e**).

polynomial of a SAT problem corresponds to Eq. (3) with $w_h = 1 \; \forall \; h$. Therefore, any SAT problem in CNF maps directly to a higher-order Ising model in which each higher-order interaction represents a clause. The order of an interaction corresponds to the size of the corresponding clause.

## Model scaling of higher-order and traditional Ising models

Quadratization of higher-order interactions introduces auxiliary variables and adds second-order interactions (XOR example in Fig. 1), thereby potentially increasing the total resources required by the corresponding Ising machine. To quantify this effect, Fig. 2 compares the resource use of higher-order models versus second-order models on $k$SAT benchmarks[44,45]. $k$SAT is a SAT problem where each clause involves maximally $k$ variables. Quadratization of $k$SAT proceeds first by reducing a $k$SAT problem to 3SAT for $k > 3$, which can always be done[46], and then quadratization of the 3SAT problem. We use the D-Wave Ocean software package for quadratization (Methods, Excess resource use by different quadratization methods), which accepts the minimum classical energy gap, $\Delta E_{min}$, as an input parameter. $\Delta E_{min}$ is the difference in energy between satisfied states and the lowest energy unsatisfied state. The choice of minimum energy gap value influences the annealing time in quantum adiabatic annealing[3] and the state acceptance probability in simulated annealing[47]. Increasing the minimum energy gap for an Ising machine may improve the optimization, however, it tends to increase the number of auxiliary variables and interactions required (Methods, Excess resource use by different quadratization methods). We compare higher-order to second-order models in terms of the number of variables in the energy function and the number of connections needed to implement all interactions. We consider second-order models with different minimum energy gap values. Nearby values of $\Delta E_{min}$ result in the same quadratization, therefore, we investigate $\Delta E_{min}$ settings of 1, 5, 10, and 13 where 1, 2, 3, and 5 auxiliary variables are introduced per clause, respectively. In addition, we found that the method used to perform quadratization increases the required precision or resolution of coupling coefficients from one bit for factored higher-order Ising models to at most six bits. This is another significant difference in resource requirements, as hardware typically offer limited resolution precision for representing interactions[31].

To compare the resource use of interactions of different orders we consider the number of connections between nodes that are required for their implementation. The required number of connections depends on the way a higher-order interaction is implemented, here we compare two methods of implementation. The first method is bidirectional connections between all variables participating in the higher-order interaction—a $k$th-order interaction requires $k(k-1)$ connections (Fig. 2c). The second method uses an intermediate computational node that receives input from all other variables participating in the interaction and sends output back to all other variables—a $k$th-order interaction requires $2k$ connections (Fig. 2e).

Our comparison shows that second-order models based on quadratization of higher-order models require a much greater number of variables and connections compared to higher-order models for $k$SAT benchmarks Fig. 2. In particular, second-order models require three orders of magnitude more variables and one order of magnitude more connections compared to higher-order models. In addition, the number of variables obtained from the D-Wave Ocean software package for $\Delta E_{min} = 1$ is the same as another method of quadratization based on a circuit decomposition of SAT clauses[48] which introduces one auxiliary variable per clause (Methods, Excess resource use by different quadratization methods).

## Solving SAT problems with a higher-order oscillator Ising machine

We compared higher-order and second-order oscillator Ising machines in their ability to solve $k$SAT problems from a benchmark dataset[44,45]. In these $k$SAT problems, the number of clauses scales linearly with the number of variables and solutions are hard to find because the satisfying states occupy only a tiny fraction of the state space. For additional details about the $k$SAT benchmark dataset, see Methods section, Resource calculations.

Our networks for implementing Ising machines use the Hopf oscillator, an oscillator model that includes amplitude dynamics. Such network models reflect the behavior of oscillator hardware more accurately than models with fixed oscillator amplitudes such as Kuramoto models[34]. In addition, our choice is motivated by simulation experiments indicating that Hopf oscillators with dynamic amplitudes provide far better solutions to the $k$SAT benchmark problems than Kuramoto networks (Fig. S2 of the Supplementary Information file). Following previous work on oscillator Ising machines with the Kuramoto model[10] our model uses sub-harmonic injection locking[10,37]. In the resulting higher-order oscillator Ising machine, the amplitude and phase of an oscillator are described by a complex variable $z_i \in \mathbb{C}$, which evolves according to:

$$\dot{z}_i(t) = f(z_i(t)) - r_i(t) \frac{\partial E(g(\mathbf{z}(t)))}{\partial z_i} + q_i(t)\, l(z_i(t)). \qquad (5)$$

On the right-hand side, $f(z_i)$ (Eq. (15) in Methods, Oscillator model and simulation details) is the local oscillator dynamics, and $\frac{\partial E(g(\mathbf{z}(t)))}{\partial z_i}$ the partial derivative of the Ising energy with respect to oscillator $z_i$, with time-dependent coupling coefficient $r_i(t)$, and optional element-wise non-linearity, $g(\mathbf{z}(t)) = \mathbf{z}(t)/|\mathbf{z}(t)|$ for normalizing the amplitude of each oscillator. Further, $l(z_i) = \bar{z}_i$ is the phase quantization signal driving the phase of oscillator $z_i$ to discrete states, with time-dependent "annealing" coefficient, $q_i(t)$. The phase quantization signal is equivalent to sub-harmonic injection locking (Methods, Oscillator model and simulation details).

Higher-order oscillator Ising machines achieve better solutions than second-order oscillator Ising machines on all 3SAT benchmark problems, as measured by mean energy at the solution points (Fig. 3a). Only for the smallest problem instances (20 variables), the difference is small. For larger problems, a substantial gap in energy appears and increases with problem size. Interestingly, even second-order oscillator Ising machines with large minimum energy gaps and, correspondingly, high resource use cannot close the performance gap to higher-order oscillator Ising machines. The performance gap amounts to about 0.75 percent of constraints satisfied for the large 3SAT problems (Fig. 3b). Finding optimal solutions, i.e., states that satisfy all the constraints, is a hard problem as there could be very few satisfying states in the entire state space. Nevertheless, for larger problems of the 3SAT benchmarks, higher-order oscillator Ising machines tend to find solutions that satisfy all constraints with greater probability than the second-order oscillator Ising machines, Fig. 3c. In fact, the higher-order oscillator Ising machine is the first reported Ising machine to find satisfiable solutions to the largest 3SAT problems (250 variables) since the previous efforts with second-order Ising machines have been unable to find solutions satisfying all clauses[48], note the missing bars in Fig. 3c.

Annealing typically improves the quality of solutions found by Ising machines[20,21,29,48]. In both our higher-order oscillator Ising machine and existing second-order oscillator Ising machines, a process analogous to adiabatic and simulated annealing is achieved by gradually increasing the coefficient in the sub-harmonic injection locking term, $q_i$[10]. We investigated linear annealing schedules with different duration, measured by the number of cycles of the resonant frequencies of the oscillators. The percentage of constraints satisfied at the end of the annealing schedule improves with the duration of the annealing schedule (Fig. 3d). The time-to-solution for reaching a fixed target of 95% of constraints satisfied (TTS$_{95}$) scales linearly with the slope in the annealing schedule (Fig. 3e). For large slopes, the TTS$_{95}$

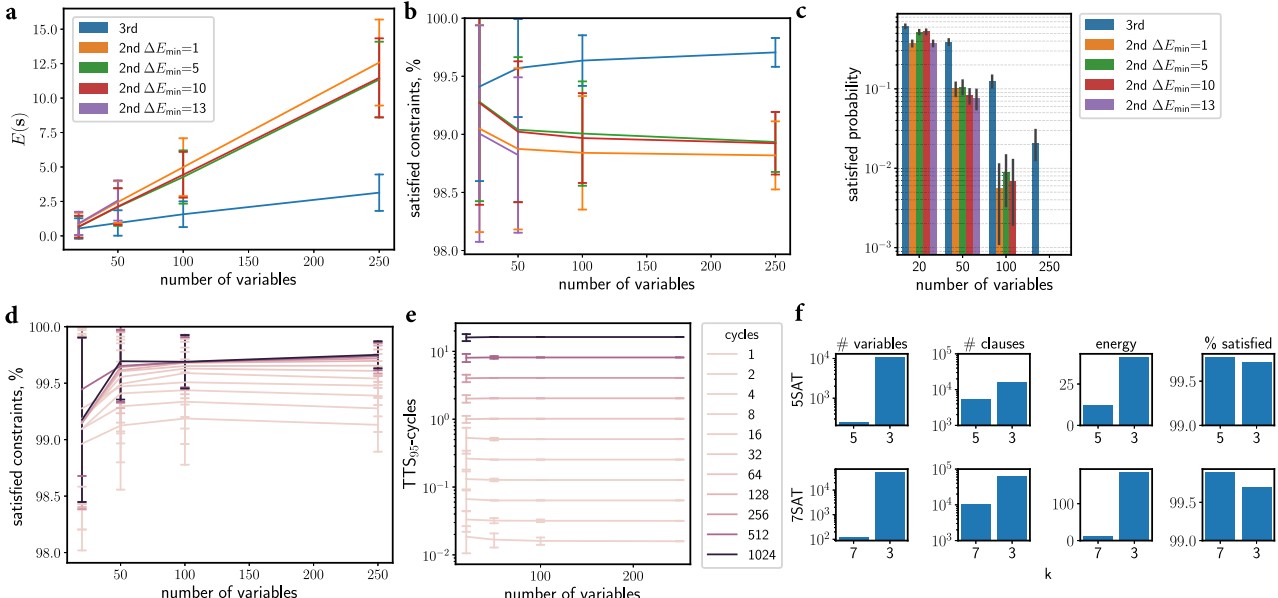

**Fig. 3 | Second-order versus higher-order networks when solving kSAT problems. a** The mean higher-order energy at the end of the simulation is plotted against the number of problem variables for hard instances of 3SAT problems. As the problem size increases, the difference in energy between the second-order oscillator Ising machines and the higher-order oscillator Ising machines increases. **b** The mean percent of constraints satisfied at the end of simulation versus the problem size for 3SAT problems. **c** The probability of satisfying all constraints for different problem sizes and models for 3SAT problems. **d** The mean percent of constraints satisfied at the end of simulation versus the problem size for higher-order oscillator Ising machines for 3SAT problems. **e** The mean time to satisfy 95% of constraints for higher-order Ising machines for 3SAT problems. **d, e** Lines indicate different linear annealing schedules for the sub-harmonic injection locking coefficients, $q_i$. In all plots, error bars represent the sample standard deviation computed over problem instances and trial simulations. **f** Comparing resources and solutions of 5SAT and 7SAT problems to their 3SAT reductions. Reducing kSAT problems to 3SAT for $k > 3$ increases the number of variables and connections (left two columns). The 5th-order and 7th-order Ising machines find lower energy states corresponding to a greater fraction of constraints satisfied compared to the 3rd-order Ising machine (right two columns).

can be a fraction of a cycle, consistent with previous findings that oscillator Ising machines rapidly find low energy states[29]. In fact, higher-order oscillator Ising machines can satisfy more than 95% in less than one cycle for all problems (for comparisons of TTS values for higher target percentages, see Fig. S1 of the Supplementary Information file).

Many studies on solving kSAT problems for $k > 3$, first use an efficient method for reducing the problem to 3SAT[46] and then focus on solving the resulting 3SAT problem. Here, we use a benchmark dataset of 5SAT and 7SAT problems[49] to assess this strategy for the higher-order oscillator Ising machine in terms of resource efficiency and solution quality (Methods, Method for reducing kSAT to 3SAT). First, we find that the reduction to 3SAT increases the number of problem variables by one or two orders of magnitude, and there is approximately a 3 and 6 times increase in the number of clauses for 5SAT and 7SAT, respectively (left two columns in Fig. 3f). Second, we observe that the direct solution of the 5SAT and 7SAT problems satisfy a greater fraction of constraints compared to solutions of corresponding 3SAT reductions (right column in Fig. 3f). It would be interesting to compare the 5th- and 7th-order oscillator Ising machines to second-order oscillator Ising machines but we were unable to test second-order oscillator Ising machines on these problems due to the large number of auxiliary variables introduced via quadratization.

### Hardware implementations of high-order oscillator Ising machines

The results presented so far suggest that higher-order oscillator Ising machines may have computational advantages over current hardware, and extending hardware implementations of oscillator Ising machines beyond second-order interactions is promising. Computing with higher-order interactions requires a state variable to form and

accumulate the partial derivatives of all terms in the total energy it participates in. Depending on the formulation of the total energy, individual terms can pertain to individual higher-order interactions $J_{i_1 \dots i_k}^{(k)} ; k > 2$ as in (4), or pertain to factored higher-order interactions representing constraints in the optimization problem (3). Regardless of which decomposition of the total energy is used, computing the derivative of an individual energy term will require information from more than one other state variable. Here we considered two options to solve this communication problem, though, others are possible. In the first option the state of each variable in a term is sent to every other variable node involved in the term, Fig. 2c. Conversely, in the second option, the states of variables participating in an energy term are first sent to an additional node in which the partial derivatives are computed. The result is communicated back to the variable nodes where the partial derivatives are accumulated Fig. 2e—for details, see Methods section, Computing derivatives of the higher-order Ising energy function.

## Discussion

Much of the existing literature on optimization with Ising machines have focused on second-order Ising networks. Such models were first proposed in ref. 2 for solving constraint satisfaction problems. The authors in ref. 2 originally proposed mapping a SAT problem to a higher-order polynomial but then applied quadratization to map to a second-order Ising model. Our first contribution is to directly compare the resource use of second- and higher-order Ising models for solving SAT problems. Defying common intuition, the comparison reveals that higher-order Ising machines are more resource-efficient than second-order Ising machines for solving large combinatorial optimization problems. The resource efficiency of higher-order models results from the fact that no auxiliary variables are required and many

combinatorial optimization problems map to polynomials which correspond to a very sparse higher-order interaction graph. Thus, the savings in higher-order models are in the number of Ising variables, as well as in the number of connections (Fig. 2).

Our second contribution is to build a resource-efficient higher-order Ising machine with coupled oscillators and test it on benchmark datasets of SAT problems. Motivated by other recent work[21,41,42,50], we investigated the implementation of a higher-order oscillator Ising machine in a coupled oscillator network. Our model resembles the one in ref. 50, but still differs in several ways. First, we use Hopf oscillators which include amplitude dynamics and capture the dynamics of oscillator hardware[34] more closely than the oscillators modeled by the Kuramoto model in ref. 50. Second, we introduce a form of annealing, specifically, the gradual increase of the sub-harmonic injection locking coefficient following a linear annealing schedule. Third, our model uses the simplest energy function resulting from the mapping method in ref. 2 (Eqs. (1) or (2)), a sum of all constraint terms where each constraint term is a product of binary values. In principle, the mapping method specifies an entire family of valid energy functions in which the products in the constraint energy are raised by any positive exponent before summing them. For example, in[4,50] the constraint terms are squared before summing. Our model choice results in gradient computations with the lowest possible complexity, and, moreover, achieves better solutions on the benchmark problems than a model with squared constraints (Methods, Comparing higher-order constraint energy functions with different exponents).

Higher-order oscillator Ising machines converge to optimal or near-optimal solutions in very few cycles, and importantly, convergence time does not increase with problem size (Fig. 3e). In some practical cases, solutions are reached in less than one cycle. Further, higher-order oscillator Ising machines outperform second-order Ising machines in solution quality and in some cases find optimal solutions to Boolean constraint satisfaction problems. To our knowledge, this study is the first to report an Ising machine that finds optimal satisfiable solutions for the large 3SAT problems in the benchmark dataset (Fig. 3c).

It has to be emphasized that our study focuses on optimization methods with a basic Ising model whose only dynamic variables are the spin variables. These methods are extremely fast and resource-efficient, but they sometimes find only near-optimal solutions. Another type of Ising machine with higher-order interactions implements the Lagrange method[4,51,52], consisting of two types of dynamic variables, spin variables and Lagrange multipliers. In these models, each constraint term in the objective function is multiplied with a nonnegative variable, the Lagrange multiplier. If a constraint is unsatisfied, the corresponding Lagrange multiplier grows dynamically, until the constraint is satisfied[4,51,52]. In theory, the Lagrange models can find optimal solutions in polynomial time but the multipliers can grow exponentially large as a function of time[4]. Further, the time to solution in Lagrange models increases with problem size[4,52]. The systematic comparison of Lagrange methods with higher-order versus second-order interactions is an interesting topic for future research.

The reported benefits of higher-order Ising machines, and higher-order oscillator Ising machines, in particular, are practically relevant because today many technologies exist for their realization. For example, higher-order interactions require the multiplication of the variables involved in the interaction. The multiplication of coupled electrical ring oscillator voltages can be implemented in the analog domain using existing CMOS technologies[33]. Further, the $k$-th-order interactions of electrical oscillators can be implemented in $\log_2(k)$ stages using a cascade of two-input multipliers or in one stage by a sequence consisting of element-wise log transform, summation, and anti-log transform. Another interesting technology is translinear electronic circuits which make use of the translinear principle[32]. Finally, existing methods for implementing real-valued analog higher-order interactions[53] may be modified for use in higher-order oscillator Ising machines.

## Methods

### Mapping optimization problems to higher-order Ising models

To express the objective function of a combinatorial optimization problem as the energy function of an Ising model, binary variables in the optimization problem must be mapped to the spins of the Ising model. In this study, the transformation between problem variables, $x_i \in \{0, 1\}$, and spins, $s_i \in \{-1, 1\}$, uses the standard transformation: $s_i = 2x_i - 1$.

### Equivalence of higher-order Ising energy formulations

It is easy to see that Eqs. (1) and (2) are equivalent. For constraint $h$, the corresponding sets $\bar{\mathbf{C}}_h$ or $\mathbf{C}_h$ partition the state space. Therefore, any state, $s$, is an element of one of the two sets and we have:

$$\sum_{\mathbf{c} \in \bar{\mathbf{C}}_h} \prod_{i=1}^{k} (1 + c_i s_i)/2 = 1 - \sum_{\mathbf{c} \in \mathbf{C}_h} \prod_{i=1}^{k} (1 + c_i s_i)/2,$$

with Eq. (1) on the LHS and Eq. (2) on the RHS. The product terms evaluate to 1 when $\mathbf{s} = \mathbf{c}$ and 0 otherwise. If $\mathbf{s} \in \bar{\mathbf{C}}_h$, both sides equal 1, if $\mathbf{s} \in \mathbf{C}_h$, both sides equal 0. Therefore, Eqs. (1) and (2) represent the same objective function and can be used interchangeably.

### Computing derivatives of the higher-order Ising energy function

If the number of terms in the sum in equation (1) is small, it is efficient to compute the partial derivative of the total energy as a sum of derivatives of individual unsatisfied constraint terms:

$$\frac{\partial E_h(\mathbf{z})}{\partial z_i} = -\sum_{c \in \bar{C}_h} c_i \prod_{j \neq i} (1 + c_j z_j)/2. \tag{6}$$

Conversely, if the number of terms in the sum of (2) is small, it is efficient to compute the partial derivative of the total energy as a sum of derivatives of individual satisfied constraint terms:

$$\frac{\partial E_h(\mathbf{z})}{\partial z_i} = \sum_{c \in C_h} c_i \prod_{j \neq i} (1 + c_j z_j)/2 \tag{7}$$

Alternatively, if the number of terms in the energy expansion (4) is small, it is efficient to compute the partial derivatives of individual $J_{i_1 \cdots i_l}^{(l)}$-terms which a variable interacts with:

$$\frac{\partial E(\mathbf{z})}{\partial z_i} = \sum_{\{J_{i_1 \cdots i_l}^{(l)}\}} \frac{\partial}{\partial z_i} E_{J_{i_1 \cdots i_l}^{(l)}}(\mathbf{z}) = \sum_{\{J_{i_1 \cdots i_l}^{(l)}\} : i \in \{i_1 \cdots i_l\}} J_{i_1 \cdots i_l}^{(l)} \prod_{v \in \{i_1 \cdots i_l\} \setminus \{i\}} z_v. \tag{8}$$

### Method for reducing $k$SAT to 3SAT

In this study, we also investigate a polynomial-time method[46] to reduce $k$SAT to 3SAT when $k > 3$. The method works as follows. Let $\wedge$, $\vee$, and $\sim$ denote the logical OR, AND, and NOT operations, respectively. Consider a clause with 5 binary variables, $(x_1 \vee x_2 \vee x_3 \vee x_4 \vee x_5)$. Introduce auxiliary variables $y_1$ and $y_2$. Introduce new clauses and insert auxiliary variables as:

$$(x_1 \vee x_2 \vee \tilde{y}_1) \wedge (x_3 \vee x_4 \vee \tilde{y}_2) \wedge (y_1 \vee y_2 \vee x_5).$$

The problem is 3SAT as no clause has greater than three variables. Reducing one 5SAT clause to 3SAT form results in 3 clauses and 7 variables.

Consider a clause with seven variables, $(x_1 \vee x_2 \vee x_3 \vee x_4 \vee x_5 \vee x_6 \vee x_7)$. Introduce auxiliary variables $y_1, y_2$, and $y_3$. Introduce

new clauses and insert auxiliary variables:

$$(x_1 \lor x_2 \lor \tilde{y}_1) \land (x_3 \lor x_4 \lor \tilde{y}_2) \land (x_5 \lor x_6 \lor \tilde{y}_3) \land (x_7 \lor y_1 \lor y_2 \lor y_3).$$

The last clause contains four variables so it has to be reduced further. Introduce auxiliary variables $l_1$ and $l_2$. Introduce new clauses and insert auxiliary variables:

$$(x_1 \lor x_2 \lor \tilde{y}_1) \land (x_3 \lor x_4 \lor \tilde{y}_2) \land (x_5 \lor x_6 \lor \tilde{y}_3) \land (x_7 \lor y_1 \lor \tilde{l}_1) \land (y_2 \lor y_3 \lor \tilde{l}_2) \land (l_1 \lor l_2).$$

The problem is 3SAT as no clause has greater than three variables. Reducing one 7SAT clause to 3SAT results in six clauses and 12 variables.

### Excess resource use by different quadratization methods

Numerous quadratization methods have been proposed for reducing objectives with higher-order interactions to energy functions of second-order Ising energies[2,17,22,24,48]. In general, the number of auxiliary variables introduced by quadratization depends on the particular combinatorial optimization problem and the method of quadratization. In this study, quadratization was performed with the D-Wave Ocean software package (https://docs.ocean.dwavesys.com/en/stable). With the quadratization method in D-Wave Ocean one can adjust the minimum energy gap, $\Delta E_{min}$, for controlling the tradeoff between excess resource use and computation performance of the resulting second-order Ising machine.

With the parameter choice of $\Delta E_{min} = 1$, one auxiliary variable is introduced per 3SAT clause, the same number as with some other quadratization methods[48]. Thus, the excess resource use of quadratization we report for this parameter choice generalizes to other methods in the literature. In addition, we also assess the resource use with parameter settings of $\Delta E_{min} > 1$ in D-Wave Ocean. These results are specific to the D-Wave Ocean quadratization method, but informative for exploring whether increased excess resource use could potentially close the performance gap between second-order and higher-order oscillator networks.

### Benchmark datasets

We assess the performance of higher-order Ising machines on Boolean satisfiability ($k$-satisfiability, $k$SAT) problems, a well-known class of hard combinatorial optimization problems. Specifically, the 3SAT problems used in our experiments were obtained from the SATLIB collection[44] (https://www.cs.ubc.ca/~hoos/SATLIB/benchm.html). We selected instances of sizes 20, 50, 100, and 250 variables. The first sixteen instances were selected from each problem size to run the simulations. The dynamic variables in the oscillator networks were randomly initialized for each trial simulation. 64 trial simulations were performed for each instance.

To demonstrate the performance of higher-order Ising machines on 5SAT and 7SAT problems, we selected an instance of each problem from the 2018 SAT Competition[49] (https://satcompetition.github.io/2018/). The 5SAT and 7SAT problems were also reduced to 3SAT using the method described in the Section "Method for reducing $k$SAT to 3SAT".

### Resource calculations

We use benchmark $k$SAT problems to demonstrate the advantages of higher-order Ising models compared to second-order Ising models in terms of resource utilization. The chosen benchmark problems are considered hard because they possess only a few satisfying states in a vast state space. Such hard problems are found amongst random instances that are sampled with a specific clause-to-variable ratio, $\alpha_{clause:v}^k$ for each value of $k$[49]. The results in Figs. 2 and 3 used problems with clause sizes of $k$ equal to 3, 5, and 7, and $\alpha_{clause:v}^k$ equal to 4.267, 21.117, and 87.79, respectively.

In order for a problem to be solved by a second-order Ising machine the $k$SAT problems must first be reduced to 3SAT for $k > 3$, which introduces auxiliary variables and additional clauses. Let $N_v^{k,k}$ be the number of variables for a $k$SAT problem represented with $k$th-order interactions. Then,

$$N_{clause}^{k,k} = \alpha_{clause:v}^k N_v^{k,k} \tag{9}$$

is the number of $k$SAT clauses. After reducing $k$SAT to 3SAT, the number of third-order clauses is

$$N_{clause}^{3,k} = N_{clause}^{k,k} \alpha_{clause:clause}^{3:k}, \tag{10}$$

where $\alpha_{clause:clause}^{3:k}$ is the clause-to-clause ratio stemming from reducing $k$SAT to 3SAT. The number of variables in the reduced 3SAT problem is

$$N_v^{3,k} = N_v^{k,k} + N_{clause}^{k,k} N_{v:clause}^{3:k}, \tag{11}$$

where $N_{v:clause}^{3:k}$ is the expected number of auxiliary variables introduced when reducing a $k$SAT clause to a 3SAT clause. For second-order Ising machines, the 3SAT problem needs to be reduced further to a second-order MAXSAT problem. The number of variables in the second-order Ising model is

$$N_v^{2,k} = N_v^{3,k} + N_{clause}^{3,k} N_{v:clause}^{2:3}, \tag{12}$$

where $N_{v:clause}^{2:3}$ is the expected number of auxiliary variables introduced during quadratization when reducing a 3SAT clause to second-order interactions.

For a $k$SAT problem implemented in an $l$th-order interactions Ising model, the number of connections is:

$$N_{conn}^{l,k} = N_{clause}^{l,k} N_{conn:clause}^l. \tag{13}$$

Here, $N_{conn:clause}^l$ is the number of connections for an $l$th-order clause, which depends on the method for implementing higher-order interactions. Two methods are compared. The first method uses $l(l-1)$ connections and the second $2l$ connections for implementing one $l$th-order interaction.

The number of connections in the second-order model is

$$N_{conn}^{2,k} = N_{clause}^{3,k} N_{conn:clause}^{2:3}, \tag{14}$$

where $N_{conn:clause}^{2:3}$ is the number of second-order connections required to implement the 3SAT clause. The number of second-order connections depends on the number of auxiliary variables introduced during quadratization, which, in turn, depends on $\Delta E_{min}$. For a 3SAT clause implemented by $n$ variables with second-order interactions, $n(n-1)$ connections are required.

### Oscillator model and simulation details

In higher-order oscillator Ising machines, each oscillator is represented by the complex Van der Pol or Hopf oscillator as described in Eq. (15):

$$f(z_i) = (\lambda_i + i\omega_i)z_i + \rho_i z_i |z_i|^2. \tag{15}$$

Here, $\omega_i$ is the center frequency for the $i$th oscillator, $\lambda_i$ is a parameter determining the oscillator quality, and $\rho_i$ controls the degree of nonlinearity.

In our simulations, the network coupling, $r_i(t)$, was the same for all oscillators and was held constant for the duration of the simulation. The center frequency was held constant at zero for all oscillators, $\omega_i = 0 \ \forall \ i$. The parameters $\lambda_i$ and $\rho_i$ were set to produce limit-cycle oscillations with unit amplitude. We used a linear annealing

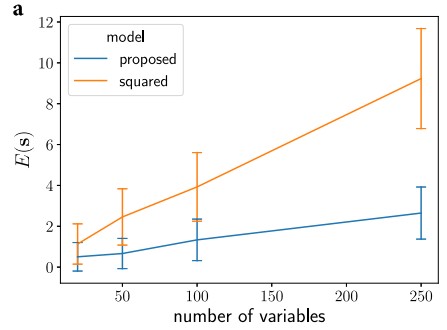
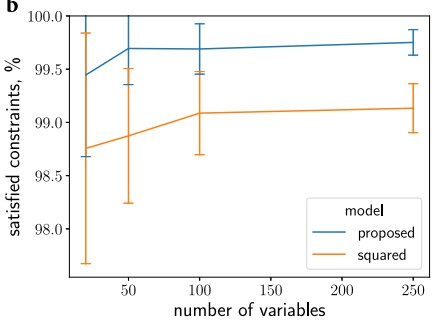

**Fig. 4 | Comparing higher-order energy functions.** Final energy on benchmarks 3SAT problems for the method proposed in this paper using Eq. (16) and the method using the square of the constraint energy (17) as in refs. 4,50. **a** Energy versus the number of variables for 3SAT problems. **b** The percent of constraints satisfied versus the number of variables for 3SAT problems.

schedule, $q_i(t) = q_{max} \frac{t}{t_{end}}$. The phase quantization signal, $l(z_i)$ is equivalent to sub-harmonic injection locking. We show this by representing each oscillator, $z_i$, with a real and imaginary part $(a_i + ib_i)$. By adding the conjugate of $z_i$ to the dynamics, the real part grows and the imaginary part decays to zero. The solutions to the dynamics for each uncoupled oscillator including the limit-cycle dynamics, $f(z_i)$, are $a_i = \sqrt{(h_i + \lambda_i)/\rho_i}$

The results reported in Fig. 3 were obtained using a parameter search to find the optimal values of $\lambda$, $\rho$, $q_{max}$, $t_{end}$, and $r$ individually for both the higher-order and second-order models. The best candidates were selected based on the lowest mean energy and the greatest mean probability of satisfying problem instances. The mean energy and percent of constraints satisfied were computed based on the final state of the network after simulation. The mean was computed across random network initializations for all trail simulations across problem instances within each problem size. Tables S1–S3 in the Supplementary Information file contain the 10 best parameter configurations for each problem size for the higher-order and the second-order models. The error bars in Fig. 3 represent the sample standard deviation. Integration of the dynamical system was performed using an adaptive step-size RK4/5 method (https://github.com/google/jax). The computer code used to produce the results reported in this study is available online at https://github.com/connorbybee/hoim.

### Comparing higher-order constraint energy functions with different exponents

For a $k$SAT problem, the objective for clause $h$ in our method (1) simplifies to:

$$E_h(\mathbf{s}) = \prod_{i=1}^{k} (1 - c_i s_i)/2. \quad (16)$$

with $c_i = 1$ if a literal is TRUE and $c_i = -1$ if a literal is FALSE. Since $E_h(\mathbf{s})$ evaluates to either one or zero for all bipolar state vectors, $\mathbf{s}$, an obvious generalization of the clause objective (16) is to exponentiate the RHS by a positive number. In[4,50], the objective of $k$SAT problems with a higher-order energy function of this type was proposed, with the specific setting of the exponent set to a value of two:

$$E_h(\mathbf{s}) = \left( \prod_{i=1}^{k} (1 - c_i s_i)/2 \right)^2. \quad (17)$$

We compared the solution quality of higher-order oscillator Ising machines implementing objective (16) vs. (17). Our experiments included parameter optimization for each method, as described above (Oscillator model and simulation details). Tables S1–S3 of the

Supplementary Information file contain the 10 best parameter configurations for each problem size. Figure 4 shows that networks based on (17) obtain worse solutions (with greater energies) and satisfy only a smaller percentage of constraints on benchmark 3SAT problems[44] compared to our method. The systematic analysis of exponent settings in the generalization of our method is left to future research.

## Data availability
The data that support the plots within this study and other findings can be generated using the available code and data available in the Supplementary Information file.

## Code availability
The computer code used to produce the results reported in this study is available online at https://github.com/connorbybee/hoim[54].

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

## Acknowledgements

C.B. acknowledges support from the National Science Foundation (NSF) through an NSF Graduate Research Fellowships Program (GRFP) fellowship (DGE 1752814) and an Intel Corporation research grant. D.K. acknowledges support from the European Union's Horizon 2020 research and innovation programme under the Marie Skłodowska-Curie grant agreement No. 839179. F.T.S. was supported by NSF Grant IIS1718991 and NIH Grant 1R01EB026955. B.A.O. was supported by NSF EAGER grant 2147640.

## Author contributions

C.B., D.K., D.E.N., A.K., B.A.O., and F.T.S. participated in discussions shaping the ideas and defining the research questions in this study. C.B. proposed the mathematical formulation of higher-order oscillator Ising machines and implemented the simulation experiments. C.B., D.K., and F.T.S. wrote the manuscript with input from all authors.

## Competing interests

The authors declare no competing interests.
