## [Peer Review File · Nature Communications]

REVIEWER COMMENTS

Reviewer #1 (Remarks to the Author):

The manuscript entitled "Efficient Optimization with Higher-Order Ising Machines" by Fredrich T. Sommer et al. has presented the results of a simulation study of an Ising machine implementing the higher-order interaction present in the constrained satisfiability problems with keeping its intact form without reducing the polynomial order. By using the proposed method, the authors have shown that the resource efficiency for the calculation highly increases, which is thought to be a big achievement in the field of the algorithm of the Ising machine. Nevertheless, I feel some deficiencies that should be supplemented to make the results more concrete.

1. The procedure of the calculations is too abstract. For example, I think that the description of the procedure for obtaining the results in Fig. 2 and Fig. 3 is not enough, preventing the reviewer from judging the correctness or the fitness of the models and the assumptions. I believe that supplementing that description of the simulation will make the results more convincing.

2. I'm curious about the method of implementing the higher-order interaction in the hardware Ising machine. I understand the direct capacitive- or resistive-coupling between electrical oscillators gives the second-order interaction in the energy function. However, I can't imagine how to couple oscillators for implementing the higher-order or the indirect interactions. I see that the authors say some things on this issue in the last paragraph in the Discussion section, but a more concrete figure with the connection diagram will be more helpful.

3. In Figure 3f, a typo is detected. "Conjuntive" should be corrected to "Conjunctive".

Reviewer #2 (Remarks to the Author):

The manuscript examines the suitability of Ising machines in providing solutions to Boolean satisfiability problems (SAT), and its main claim is that higher-order Ising models exhibit superior efficiency in performing such tasks compared to the more standard second-order Ising models. I find this central claim both compellingly supported by the authors' numerical results, as well as significant in terms of scientific impact. I particularly appreciated the pedagogical clarity of the figures in distilling complex ideas in a way that even the non-specialist can follow (Fig. 1 is a great example of this).

One question that emerged for me was: are higher-order Ising interactions physically realizable? In the original magnetic context, one usually only sees pairwise terms in the Hamiltonian. The authors address this question later on in the Discussion section, where potential implementations of beyond-pairwise interactions are mentioned. These could involve, for instance, analog multipliers (CMOS) of electronic signals deriving from electronic oscillators (if I understand correctly, also from Ref.49 & 50). In other physical contexts, however, such clever "post-processing" of signals might not be an option. As a point of curiosity, I wonder whether there are examples of higher-order Ising terms coming out of the interaction physics itself, in a sense that the (nonlinear) physics itself accomplishes the desired dynamics. (It is alright if the answer is 'no'.)

Finally, one suggestion would be to add a bit more motivation for Eq.(4) as the evolution equations of choice. The authors do provide additional information in Methods 4.7 but some elucidation of the nature of the coupling/annealing would be helpful (in the results section). Has this equation been used in other published works on Hopf oscillators?

In conclusion, the manuscript is well written, reports results that are noteworthy and significant, and should be considered for publication.

Reviewer #3 (Remarks to the Author):

Referee report on the manuscript entitled "Efficient Optimization with Higher-Order Ising Machines" by Connor Bybee, et al.

The authors study the performance of high-order Ising Machine in the solution of k-SAT problems from a theoretical point of view. The topic of the work is of interest and timely, I have to say that the manuscript is well written, and the motivation and results are well described. However, while I think this work should be published in some form before to recommend it for NCOMM the following points should be addressed.

The authors present the TTS 95%, this is a nice metrics but currently the main problem an Ising machine has to solve is moving forward 95 while maintaining good performance, the TTS 96%, 97%, 98%, 99% and 100% of the high order IM should be presented and compared with 2nd order IM and potentially with other approaches.

The information about the number of clauses is a bit confusing when discussing Fig. 3. The text should include those information.

The model used by the authors is more general of the Kuramoto model, but I'm not sure the performance will be better. Why the coupling with power should give better results? I agree that this model is close to the experimental behaviour of the oscillator but for a theoretical point of view I do not see any clear advantage. The effect of this coupling should be investigated in deep to understand potential advantages and a clear comparison should be shown.

An important question to address in the comparison between second order and high order IMs is the annealing process. I understand that the authors used a similar annealing scheme for both IMs. However, the claim in this work is pretty strong and the comparison should be with an optimized annealing for both approaches. Or at least it should be shown a comparison with different annealing approaches.

Response to the Comments on “Efficient Optimization with Higher-Order Ising Machines”

Reviewer 1

The manuscript entitled “Efficient Optimization with Higher-Order Ising Machines” by Fredrich T. Sommer et al. has presented the results of a simulation study of an Ising machine implementing the higher-order interaction present in the constrained satisfiability problems with keeping its intact form without reducing the polynomial order. By using the proposed method, the authors have shown that the resource efficiency for the calculation highly increases, which is thought to be a big achievement in the field of the algorithm of the Ising machine. Nevertheless, I feel some deficiencies that should be supplemented to make the results more concrete.

We thank the reviewer for accurately catching and summarizing the essence of our study. We also appreciate the raised points and provide our responses below.

1. **The procedure of the calculations is too abstract. For example, I think that the description of the procedure for obtaining the results in Fig. 2 and Fig. 3 is not enough, preventing the reviewer from judging the correctness or the fitness of the models and the assumptions. I believe that supplementing that description of the simulation will make the results more convincing.**

Following your suggestion, we have added a subsection “Resource calculations” in Methods that describes the procedure for calculating the number of variables and connections for higher-order and second-order Ising models:

We use benchmark k SAT problems to demonstrate the advantages of higher-order Ising models compared to second-order Ising models in terms of resource utilization. The chosen benchmark problems are random instances sampled with a specific clause-to-variable ratio, $\alpha_{\text{clause:var}}^k$ for each value of k [1]. Here, var stands for variables, clause stands for a problem clause, and k is the order of the clause, i.e., the number of variables per clause. At the chosen value of $\alpha_{\text{clause:var}}^k$, the problems exist at a transition from satisfiable and unsatisfiable. There are only a few satisfying states in vast state space. Thus, finding satisfying solutions is a hard problem. The results in Fig. 2 & 3 used problems with clause sizes of k equal to 3, 5, and 7, and $\alpha_{\text{clause:var}}^k$ equal to 4.267, 21.117, and 87.79, respectively.

In order to solve a k SAT problem with $k > 3$ by using a second-order Ising machine, the problem must first be reduced to 3SAT, which introduces auxiliary variables and additional clauses. Let $N_{\text{var}}^{k,k}$ be the number of variables for a k SAT problem represented with k th-order interactions. Here, the first index in the superscript represents the order of the reduced

problem and the second index represents the order of the original problem. Then,

$$N_{\text{clause}}^{k,k} = \alpha_{\text{clause:var}}^k N_{\text{var}}^{k,k} \quad (1)$$

is the number of k SAT clauses. After reducing k SAT to 3SAT, the number of third-order clauses is

$$N_{\text{clause}}^{3,k} = N_{\text{clause}}^{k,k} \alpha_{\text{clause:clause}}^{3:k} \quad (2)$$

where $\alpha_{\text{clause:clause}}^{3:k}$ is the clause-to-clause ratio stemming from reducing k SAT to 3SAT. $\alpha_{\text{clause:clause}}^{3:k}$ is determined by the method used for reducing k SAT to 3SAT (see Methods 5.4). The number of variables in the reduced 3SAT problem is

$$N_{\text{var}}^{3,k} = N_{\text{var}}^{k,k} + N_{\text{clause}}^{k,k} N_{\text{var:clause}}^{3:k} \quad (3)$$

where $N_{\text{var:clause}}^{3:k}$ is the expected number of auxiliary variables introduced when reducing a k SAT clause to a 3SAT clause. For second-order Ising machines, the 3SAT problem needs to be reduced further to a second-order MAXSAT problem. The number of variables in the second-order Ising model is

$$N_{\text{var}}^{2,k} = N_{\text{var}}^{3,k} + N_{\text{clause}}^{3,k} N_{\text{var:clause}}^{2:3} \quad (4)$$

where $N_{\text{var:clause}}^{2:3}$ is the expected number of auxiliary variables introduced during quadratization when reducing a 3SAT clause to second-order interactions.

For a k SAT problem implemented in an l th-order interactions Ising model, the number of connections is:

$$N_{\text{conn}}^{l,k} = N_{\text{clause}}^{l,k} N_{\text{conn:clause}}^l \quad (5)$$

Here, conn represents variable connections, clause represents the problem clause, and l is the order of the clause. The number of connections is controlled by the number of connections required for an l th-order clause, $N_{\text{conn:clause}}^l$, which depends on the method for implementing higher-order interactions. Two methods are compared. The first method uses $l(l-1)$ connections and the second $2l$ connections for implementing one l th-order interaction.

The number of connections in the second-order model is

$$N_{\text{conn}}^{2,k} = N_{\text{clause}}^{3,k} N_{\text{conn:clause}}^{2:3} \quad (6)$$

where $N_{\text{conn:clause}}^{2:3}$ is the number of second-order connections required to implement the 3SAT clause. The number of second-order connections depends on the number of auxiliary variables introduced during quadratization, which, in turn, depends on ΔE_{min} . For a 3SAT clause implemented by n variables with second-order interactions, $n(n-1)$ connections are required.

2. I'm curious about the method of implementing the higher-order interaction in the hardware Ising machine. I understand the direct capacitive- or resistive-coupling between electrical oscillators gives the second-order interaction in the energy function. However, I can't imagine how to couple oscillators for implementing the higher-order or the indirect interactions. I see that the authors say some things on this issue in the last paragraph in the Discussion section, but a more concrete figure with the connection diagram will be more helpful.

We agree that this important point deserves additional detail in the manuscript, which has also been emphasized by the first comment of Reviewer 2. As the reviewer points out, the second-order interactions can be implemented with linear elements such as capacitive or resistive couplings. In contrast, higher-order interactions require nonlinear elements such as multipliers. To address this issue, we have included a section in the Results describing how higher-order interactions might be realized:

The results presented so far suggest that higher-order oscillator Ising machines may have computational advantages over current hardware, and extending hardware implementations of oscillator Ising machines beyond second-order interactions is promising. Computing with higher-order interactions requires a state variable to form and accumulate the partial derivatives of all terms in the total energy it participates in. Depending on the formulation of the total energy, individual terms can pertain to individual higher-order interactions $J_{i_1 \dots i_k}^{(k)}$; $k > 2$ as in (4), or pertain to factored higher-order interactions representing constraints in the optimization problem (3). Regardless of which decomposition of the total energy is used, computing the derivative of an individual energy term will require information from more than one other state variable. Here we considered two options to solve this communication problem, though, others are possible. In the first option the state of each variable in a term is sent to every other variable node involved in the term, Fig. 2c. Conversely, in the second option, the states of variables participating in an energy term are first sent to an additional node in which the partial derivatives are computed. The result is communicated back to the variable nodes where the partial derivatives are accumulated Fig. 2e – for details, see Methods 4.3.

Additionally, we expanded Methods section 4.3 showing how partial derivatives can be computed for both the factored and expanded Ising energy functions.

Alternatively, if the number of terms in the energy expansion (4) is small, it is efficient to compute the partial derivatives of individual $J_{i_1 \dots i_l}^{(l)}$ -terms which a variable interacts with:

$$\frac{\partial E(\mathbf{z})}{\partial z_i} = \sum_{\{J_{i_1 \dots i_l}^{(l)}\}} \frac{\partial}{\partial z_i} E_{J_{i_1 \dots i_l}^{(l)}}(\mathbf{z}) = \sum_{\{J_{i_1 \dots i_l}^{(l)}\}; i \in \{i_1 \dots i_l\}} J_{i_1 \dots i_l}^{(l)} \prod_{v \in \{i_1 \dots i_l\} \setminus \{i\}} z_v. \quad (7)$$

3. In Figure 3f, a typo is detected. “Conjunctive” should be corrected to “Conjunctive”.

The typo has been corrected.

Reviewer 2

The manuscript examines the suitability of Ising machines in providing solutions to Boolean satisfiability problems (SAT), and its main claim is that higher-order Ising models exhibit superior efficiency in performing such tasks compared to the more standard second-order Ising models. I find this central claim both compellingly supported by the authors' numerical results, as well as significant in terms of scientific impact. I particularly appreciated the pedagogical clarity of the figures in distilling complex ideas in a way that even the non-specialist can follow (Fig. 1 is a great example of this).

First, we would like to thank the reviewer for his/her encouraging feedback. Second, we appreciate the reviewer providing the points for improving the writing and structure of the manuscript. Below, we describe our reflections on the suggestions and the corresponding actions in the revised manuscript.

1. **One question that emerged for me was: are higher-order Ising interactions physically realizable? In the original magnetic context, one usually only sees pairwise terms in the Hamiltonian. The authors address this question later on in the Discussion section, where potential implementations of beyond-pairwise interactions are mentioned. These could involve, for instance, analog multipliers (CMOS) of electronic signals deriving from electronic oscillators (if I understand correctly, also from Ref.49 & 50). In other physical contexts, however, such clever “post-processing” of signals might not be an option.**

This crucial question has also been raised in the second comment of Reviewer 1. The multiplication required to implement a higher-order interaction can be implemented with existing technologies commonly used in phase detectors and mixers. In response to this question, we have revised the last paragraph of the introduction to state this more explicitly.

Further, the multiplication and routing of electrical signals that are required to implement a k th-order interaction for an arbitrary order k can be realized with existing technologies commonly used in devices such as phase detectors and mixers [32]-[34] offering advantages over other physical systems where higher-order interactions may be limited to a specific order with limited degrees of freedom [35].

In addition, in response to this comment (and the similar comment from reviewer 1), we've included a new section in Results describing how higher-order interactions might be realized in hardware.

The results presented so far suggest that higher-order oscillator Ising machines may have computational advantages over current hardware, and extending hardware implementations of oscillator Ising machines beyond second-order interactions is promising. Computing with higher-order interactions requires a state variable to form and accumulate the partial derivatives of all terms in the total energy it participates in. Depending on the formulation of the total energy, individual terms can pertain to individual higher-order interactions $J_{i_1 \dots i_k}^{(k)}$; $k > 2$ as in (4), or pertain to factored higher-order interactions representing constraints in the optimization problem (3). Regardless of which decomposition of the total energy is used, computing the derivative of an individual energy term will require information from more than one other state variable. Here we considered two options

to solve this communication problem, though, others are possible. In the first option the state of each variable in a term is sent to every other variable node involved in the term, Fig. 2c. Conversely, in the second option, the states of variables participating in an energy term are first sent to an additional node in which the partial derivatives are computed. The result is communicated back to the variable nodes where the partial derivatives are accumulated Fig. 2e – for details, see Methods 4.3.

Additionally, we expanded Methods section 4.3 showing how partial derivatives can be computed for both the factored and expanded Ising energy functions.

Alternatively, if the number of terms in the energy expansion (4) is small, it is efficient to compute the partial derivatives of individual $J_{i_1 \dots i_l}^{(l)}$ -terms which a variable interacts with:

$$\frac{\partial E(\mathbf{z})}{\partial z_i} = \sum_{\{J_{i_1 \dots i_l}^{(l)}\}} \frac{\partial}{\partial z_i} E_{J_{i_1 \dots i_l}^{(l)}}(\mathbf{z}) = \sum_{\{J_{i_1 \dots i_l}^{(l)}\}: i \in \{i_1 \dots i_l\}} J_{i_1 \dots i_l}^{(l)} \prod_{v \in \{i_1 \dots i_l\} \setminus \{i\}} z_v. \quad (8)$$

2. **As a point of curiosity, I wonder whether there are examples of higher-order Ising terms coming out of the interaction physics itself, in a sense that the (nonlinear) physics itself accomplishes the desired dynamics. (It is alright if the answer is 'no'.)**

Yes, a very interesting comment. Various science areas offer examples of higher-order interactions, for example, biological neural networks [2], protein evolution [3], optical systems [4], and more. But to date, the main example of higher-order network phenomena in physics are coupled oscillators (references 100-105 in [5]). As the reviewer suggests, even conventional transistor circuits can exhibit higher-order interactions. For example, if the internal nodes of three oscillators are connected to the source, drain, and gate of a MOSFET, this would implement a third-order interaction between them. In this study, we examine oscillators coupled with analog multipliers, well-known electrical components. We agree that exploiting higher-order interactions in materials for computation is an exciting, yet currently overlooked direction for future research.

3. **Finally, one suggestion would be to add a bit more motivation for Eq.(4) as the evolution equations of choice. The authors do provide additional information in Methods 4.7 but some elucidation of the nature of the coupling/annealing would be helpful (in the results section). Has this equation been used in other published works on Hopf oscillators?**

We agree that in the original manuscript, the motivation of Eq. 4 was insufficient, which has also been indicated by Reviewer 3 in the third comment. To address this, we have performed experiments comparing our model in Eq. 4 to higher-order Kuramoto models (provided in the Supplementary Material – “Kuramoto versus Hopf oscillator models”):

In some cases, phase-reduced models of coupled oscillators, like the Kuramoto model, are sufficient to implement certain functions, e.g., associative memories [1]. Coupled oscillator models that contain both amplitude and phase dynamics exhibit richer behavior

than phase-reduced models [2]-[4]. To explore the effects of different oscillator models in combinatorial optimization, we compared a Kuramoto model with only phase dynamics and a Hopf model, that contains phase and amplitude dynamics. The Kuramoto model has one phase variable per oscillator. The energy for the h th constraint for the higher-order Kuramoto oscillator Ising machine used in this study is:

$$E_h(\phi) = \sum_{c \in \bar{C}_h} \prod_{i=1}^k \left(1 + \cos\left(\frac{\pi}{2}(1 - c_i) - \phi_i\right)\right)/2. \quad (9)$$

Here, ϕ_i is the phase of the i th oscillator. We note that Eq. (2) is similar to the energy function used in [5], the only difference is that in [5] the energy function is raised to a power of 2. The resulting phase dynamics are:

$$\dot{\phi}_i = \omega_i - r_i(t) \frac{\partial E(\phi)}{\partial \phi_i} - q_i(t) \sin(2\phi_i). \quad (10)$$

Here, ϕ_i is the phase of the i th oscillator, ω_i is the frequency of the i th oscillator, r_i is the coupling parameter for the i th oscillator, $E(\phi)$ is the full energy in Eq. (3) with the energy for each clause written according to Eq. (S2), and q_i is the sub-harmonic injection locking parameter.

In order to determine if amplitude dynamics has an effect on the set of solutions found by oscillator Ising machines, we compared the mean energies found by the Kuramoto model versus the Hopf model. The results in Fig. S2 show that the Hopf model produces solutions with lower mean energies for all problem sizes. This suggests, that amplitude dynamics can improve the optimization procedure. Though this finding is empirical and a theoretical understanding is lacking, it led us to focus on the Hopf model in the main manuscript.

Figure S2: **Quality of solutions found by Hopf versus Kuramoto oscillator networks.** The mean energy of the found solutions is plotted against the number of problem variables for benchmark 3SAT problems (Hopf model in blue, Kuramoto model in orange).

and added the following paragraph to the Results section:

Our networks for implementing Ising machines use the Hopf oscillator, an oscillator model that includes amplitude dynamics. Such network models reflect the behavior of

oscillator hardware more accurately than models with fixed oscillator amplitudes such as Kuramoto models [34]. In addition, our choice is motivated by simulation experiments indicating that Hopf oscillators with dynamic amplitudes provide far better solutions of the kSAT benchmark problems compared to Kuramoto networks (Section S2 of the Supplementary Material). Following previous work on oscillator Ising machines with the Kuramoto model [10] our model uses sub-harmonic injection locking [10], [37].

Afterword:

In conclusion, the manuscript is well written, reports results that are noteworthy and significant, and should be considered for publication.

Thank you.

Reviewer 3

The authors study the performance of high-order Ising Machine in the solution of k-SAT problems from a theoretical point of view. The topic of the work is of interest and timely, I have to say that the manuscript is well written, and the motivation and results are well described.

Thank you for this encouragement.

However, while I think this work should be published in some form before to recommend it for NCOMM the following points should be addressed.

Below, we provide our answers to the raised concerns and pointers to the corresponding edits in the revised manuscript.

1. **The authors present the TTS 95%, this is a nice metrics but currently the main problem an Ising machine has to solve is moving forward 95 while maintaining good performance, the TTS 96%, 97%, 98%, 99% and 100% of the high order IM should be presented and compared with 2nd order IM and potentially with other approaches.**

Indeed, we just chose the TTS 95% metric because other studies within the Ising machine literature have used it. In addition, TTS 95% was convenient because both higher-order and second-order Ising machines reach 95% constraints satisfied with high probability on each trial. We, nevertheless, see your point of reporting performance for other TTS values. To explore higher TTS values we had to modify the TTS definition to include an expectation of the number of runs, when in a single run a method can achieve the required percentage level of success only with a certain probability. As suggested by the reviewer, we then conducted the experiments for higher percentage levels of success. To avoid overloading the main text, we added the results of these experiments in the Supplementary Material.

The time to reach a given quality level can be compared between higher-order and second-order Ising machines. Fig. S1 presents the TTS 97%, 98%, 99%, and 100% for higher-order and second-order Ising machines, respectively. In all cases, higher-order Ising machines achieve a lower TTS where the TTS is calculated using the probability of reaching the target percentage:

$$TTS_{X\%} = \frac{\mathbb{E}[t_{X\%}]}{p_{X\%}}. \quad (11)$$

Here, $TTS_{X\%}$ is the approximate time to reach a solution that satisfies greater than or equal to $X\%$ of constraints, $\mathbb{E}[t_{X\%}]$ is the expected time to reach the target percentage for trails that successfully reach the target, and $p_{X\%}$ is the probability of satisfying greater than or equal to $X\%$ of constraints.

Figure S1: **TTS for higher-order and second-order Ising machines.** The TTS is presented for increasing solution quality thresholds for higher-order and second-order Ising machines on benchmark 3SAT problems. **a-d** TTS for higher-order Ising machines. **e-h** TTS for second-order Ising machines.

2. **The information about the number of clauses is a bit confusing when discussing Fig. 3. The text should include those information.**

To add clarity, we created a section in Methods that explains in more detail our procedure for computing the number of clauses (“Resource calculations”):

We use benchmark k SAT problems to demonstrate the advantages of higher-order Ising models compared to second-order Ising models in terms of resource utilization. The chosen benchmark problems are considered hard because they possess only a few satisfying states in a vast state space. Such hard problems are found amongst random instances that are sampled with a specific clause-to-variable ratio, $\alpha_{clause:v}^k$ for each value of k [1]. The results in Fig. 2 & 3 used problems with clause sizes of k equal to 3, 5, and 7, and $\alpha_{clause:v}^k$ equal to 4.267, 21.117, and 87.79, respectively.

In order for a problem to be solved by a second-order Ising machine the k SAT problems must first be reduced to 3SAT for $k > 3$, which introduces auxiliary variables and additional clauses. Let $N^{k,k}$ be the number of variables for a k SAT problem represented with k th-order interactions. Then,

$$N_{clause}^{k,k} = \alpha_{clause:v}^k N_v^{k,k} \quad (12)$$

is the number of k SAT clauses. After reducing k SAT to 3SAT, the number of third-order clauses is

$$N_{clause}^{3,k} = N_{clause}^{k,k} \alpha_{clause:clause}^{3:k} \quad (13)$$

where $\alpha_{clause:clause}^{3:k}$ is the clause-to-clause ratio stemming from reducing k SAT to 3SAT. The number of variables in the reduced 3SAT problem is

$$N_v^{3,k} = N_v^{k,k} + N_{clause}^{k,k} N_{v:clause}^{3:k} \quad (14)$$

where $N_{v:clause}^{3:k}$ is the expected number of auxiliary variables introduced when reducing a k SAT clause to a 3SAT clause. For second-order Ising machines, the 3SAT problem needs

to be reduced further to a second-order MAXSAT problem. The number of variables in the second-order Ising model is

$$N_v^{2,k} = N_v^{3,k} + N_{\text{clause}}^{3,k} N_{v:\text{clause}}^{2:3}, \quad (15)$$

where $N_{v:\text{clause}}^{2:3}$ is the expected number of auxiliary variables introduced during quadratization when reducing a 3SAT clause to second-order interactions.

For a k SAT problem implemented in an l th-order interactions Ising model, the number of connections is:

$$N_{\text{conn}}^{l,k} = N_{\text{clause}}^{l,k} N_{\text{conn:clause}}^l. \quad (16)$$

Here, $N_{\text{conn:clause}}^l$ is the number of connections for an l th-order clause, which depends on the method for implementing higher-order interactions. Two methods are compared. The first method uses $l(l-1)$ connections and the second $2l$ connections for implementing one l th-order interaction.

The number of connections in the second-order model is

$$N_{\text{conn}}^{2,k} = N_{\text{clause}}^{3,k} N_{\text{conn:clause}}^{2:3}, \quad (17)$$

where $N_{\text{conn:clause}}^{2:3}$ is the number of second-order connections required to implement the 3SAT clause. The number of second-order connections depends on the number of auxiliary variables introduced during quadratization, which, in turn, depends on ΔE_{\min} . For a 3SAT clause implemented by n variables with second-order interactions, $n(n-1)$ connections are required.

In addition, we added the following text to the Results section of the revised manuscript:

Additional details about the used k SAT benchmarks can be found in Methods 4.7. Here, it is worth noting that for the considered problems the number of clauses scales linearly with the number of variables and that the chosen problems are difficult to solve since the satisfying states correspond to a tiny fraction of the state space.

3. **The model used by the authors is more general of the Kuramoto model, but I'm not sure the performance will be better. Why the coupling with power should give better results? I agree that this model is close to the experimental behaviour of the oscillator but for a theoretical point of view I do not see any clear advantage. The effect of this coupling should be investigated in deep to understand potential advantages and a clear comparison should be shown.**

We agree that our motivation for using the Hopf model over simpler ones such as the Kuramoto model was insufficient in the original manuscript – it was also pointed out by Reviewer 2 in the third comment. This is, indeed, a good point and to improve the motivation for our model choice, we now added a comparison between the Kuramoto and the Hopf models. To keep the flow of the original manuscript, in the main text we refer to the new comparison results, when introducing the Hopf model, but placed the new results in the Supplementary Material in the section “Kuramoto versus Hopf oscillator models”.

In some cases, phase-reduced models of coupled oscillators, like the Kuramoto model, are sufficient to implement certain functions, e.g., associative memories [1]. Coupled oscillator models that contain both amplitude and phase dynamics exhibit richer behavior than phase-reduced models [2]-[4]. To explore the effects of different oscillator models in

combinatorial optimization, we compared a Kuramoto model with only phase dynamics and a Hopf model, that contains phase and amplitude dynamics. The Kuramoto model has one phase variable per oscillator. The energy for the h th constraint for the higher-order Kuramoto oscillator Ising machine used in this study is:

$$E_h(\phi) = \sum_{c \in \bar{C}_h} \prod_{i=1}^k \left(1 + \cos\left(\frac{\pi}{2}(1 - c_i) - \phi_i\right)\right)/2. \quad (18)$$

Here, ϕ_i is the phase of the i th oscillator. We note that Eq. (2) is similar to the energy function used in [5], the only difference is that in [5] the energy function is raised to a power of 2. The resulting phase dynamics are:

$$\dot{\phi}_i = \omega_i - r_i(t) \frac{\partial E(\phi)}{\partial \phi_i} - q_i(t) \sin(2\phi_i). \quad (19)$$

Here, ϕ_i is the phase of the i th oscillator, ω_i is the frequency of the i th oscillator, r_i is the coupling parameter for the i th oscillator, $E(\phi)$ is the full energy in Eq. (3) with the energy for each clause written according to Eq. (S2), and q_i is the sub-harmonic injection locking parameter.

In order to determine if amplitude dynamics has an effect on the set of solutions found by oscillator Ising machines, we compared the mean energies found by the Kuramoto model versus the Hopf model. The results in Fig. S2 show that the Hopf model produces solutions with lower mean energies for all problem sizes. This suggests, that amplitude dynamics can improve the optimization procedure. Though this finding is empirical and a theoretical understanding is lacking, it led us to focus on the Hopf model in the main manuscript.

Figure S2: **Quality of solutions found by Hopf versus Kuramoto oscillator networks.** The mean energy of the found solutions is plotted against the number of problem variables for benchmark 3SAT problems (Hopf model in blue, Kuramoto model in orange).

4. An important question to address in the comparison between second order and high order IMs is the annealing process. I understand that the authors used a similar annealing scheme for both IMs. However, the claim in this work is pretty strong and the comparison should be with an optimized annealing for

both approaches. Or at least it should be shown a comparison with different annealing approaches.

We agree with the point that it is important to make sure that the comparison is fair in the sense that the improvement comes from the usage of higher-order interactions rather than from the annealing scheduler or tuning of other parameters of the model. In order to address this point and to be clear that the parameters were optimized for each IM type, we added the following note in the section “Oscillator model and simulation details” in Methods when discussing the parameter search:

The results reported in Fig. 3 were obtained using a parameter search to find the optimal values of λ , ρ , q_{max} , t_{end} , and r individually for both the higher-order and second-order models.

References

- [1] M. J. Heule, “Generating the uniform random benchmarks,” *Proceedings of SAT competition*, vol. 2018, 2018.
- [2] A. Gidon, T. A. Zolnik, P. Fidzinski, F. Bolduan, A. Papoutsis, P. Poirazi, M. Holtkamp, I. Vida, and M. E. Larkum, “Dendritic action potentials and computation in human layer 2/3 cortical neurons,” *Science*, vol. 367, no. 6473, pp. 83–87, 2020.
- [3] A. Contini and G. Tiana, “A many-body term improves the accuracy of effective potentials based on protein coevolutionary data,” *The Journal of chemical physics*, vol. 143, no. 2, 07B608.1, 2015.
- [4] S. Kumar, H. Zhang, and Y.-P. Huang, “Large-scale Ising emulation with four body interaction and all-to-all connections,” *Communications Physics*, vol. 3, no. 1, pp. 1–9, 2020.
- [5] F. Battiston, E. Amico, A. Barrat, G. Bianconi, G. Ferraz de Arruda, B. Franceschiello, I. Iacopini, S. Kéfi, V. Latora, Y. Moreno, *et al.*, “The physics of higher-order interactions in complex systems,” *Nature Physics*, vol. 17, no. 10, pp. 1093–1098, 2021.

REVIEWER COMMENTS

Reviewer #1 (Remarks to the Author):

Dear authors,

I have read through the revised manuscript and their response to my questions about the original manuscript, which are now clear to me. I see that the idea of a "higher-order Ising Machine" is novel and is well supported by the mathematical formulation and the simulation results presented in the manuscript. Especially in Fig. 3 and Fig. S1, the authors compared the performance of the higher-order Ising machine with that of the conventional second-order Ising machine, showing many improvements in the former. Therefore, I believe that many experimentalists will be inspired by this work and recommend the publication of this manuscript.

Reviewer #2 (Remarks to the Author):

The authors have addressed all my comments from the first round of review, and I am happy with the changes they made to the manuscript. I support its publication.

Reviewer #3 (Remarks to the Author):

The authors addressed some of the comments.

However I'm not fully convinced about the authors response on two points.

First of all, I have not found the parameters used to perform the comparison in Supplementary Figure S2. It will be difficult to reproduce those results. I trust the authors, but it is hard to believe to have this difference in such small problems (250 variables). I understand one can ask the code to the corresponding author. However, all the details of the comparison should be included in order to be reproducible.

I'm not satisfied with the reply to the comment number 4. This is a very important aspect. How did you find the optimal values? I expect those parameters are problem dependent. I kindly ask the authors to show comparisons considering different set of parameters.

Response to the second round of comments on “Efficient Optimization with Higher-Order Ising Machines”

Reviewer 1

Dear authors, I have read through the revised manuscript and their response to my questions about the original manuscript, which are now clear to me. I see that the idea of a “higher-order Ising Machine” is novel and is well supported by the mathematical formulation and the simulation results presented in the manuscript. Especially in Fig. 3 and Fig. S1, the authors compared the performance of the higher-order Ising machine with that of the conventional second-order Ising machine, showing many improvements in the former. Therefore, I believe that many experimentalists will be inspired by this work and recommend the publication of this manuscript.

We thank the reviewer for their time reviewing the manuscript and appreciate the positive recommendation.

Reviewer 2

The authors have addressed all my comments from the first round of review, and I am happy with the changes they made to the manuscript. I support its publication.

We thank the reviewer for agreeing to review the manuscript, providing us with suggestions, and for the eventual support.

Reviewer 3

The authors addressed some of the comments. However I'm not fully convinced about the authors response on two points.

First of all, I have not found the parameters used to perform the comparison in Supplementary Figure S2. It will be difficult to reproduce those results. I trust the authors, but it is hard to believe to have this difference in such small problems (250 variables). I understand one can ask the code to the corresponding author. However, all the details of the comparison should be included in order to be reproducible.

I'm not satisfied with the reply to the comment number 4. This is a very important aspect. How did you find the optimal values? I expect those parameters are problem dependent. I kindly ask the authors to show comparisons considering different set of parameters.

We thank the reviewer for these additional points for improving the manuscript.

To address the concerns in regard to parameters used to produce Figure S2, we have included a table (Tables S1-s3; see below) with the 10 best parameter configurations for each problem size. Additionally, we have released our code on GitHub, <https://github.com/connorbybee/hoim>. This will easily allow one to reproduce the results and get access to all the technicalities of the performed experiments.

We would also like to elaborate to comment number 4, as anticipated by the reviewer, the parameters were, indeed, problem-dependent; they were found by searching over thousands of various parameter configurations. Each parameter configuration is one possible combination drawn from the set of possible parameter values. Each parameter was varied over several orders of magnitude. A simulation timeout of 1 day was used. Parameter configurations that would take longer than that were not considered. The 10 best parameter configurations are reported for each problem size and simulation type. The reported energy and fraction of instances satisfied are averaged over many samples for each problem instance and over problem instances for each problem size. The best parameter configuration was chosen for each problem size for the higher-order Hopf oscillation simulation, the 2nd-order Hopf oscillator simulation, and the higher-order Kuramoto oscillator simulation independently. For convenience, we have included the corresponding tables (Tables S1-S3) below. We hope that these tables and the accompanying text in the Supplementary Material provide the previously missing details of finding the optimal values used to obtain the reported results.

# of variables	λ	ρ	r	q_{\max}	# of cycles	energy	satisfied probability
20	1.0	-1.0	32.0	32.0	512	0.54	0.61
			64.0	64.0	512	0.67	0.33
	2.0	-2.0	32.0	32.0	1024	0.75	0.38
				512	0.76	0.38	
			16.0	16.0	512	0.77	0.38
				1024	0.77	0.38	
	4.0	-4.0	1.0	1.0	1024	0.77	0.38
	2.0	-2.0	16.0	16.0	128	0.77	0.38
	1.0	-1.0	1.0	1.0	1024	0.78	0.38
				8.0	8.0	256	0.78
50	1.0	-1.0	32.0	32.0	1024	0.71	0.44
			16.0	16.0	1024	0.78	0.38
	2.0	-2.0	32.0	32.0	512	0.78	0.38
			1.0	-1.0	8.0	8.0	512
	2.0	-2.0	16.0	16.0	256	0.82	0.34
				32.0	32.0	1024	0.83
	1.0	-1.0	16.0	16.0	512	0.86	0.37
				32.0	32.0	256	0.86
	2.0	-2.0	32.0	32.0	256	0.86	0.37
				128	0.88	0.33	
100	1.0	-1.0	32.0	32.0	1024	1.47	0.19
			8.0	8.0	512	1.48	0.17
	2.0	-2.0	32.0	32.0	256	1.48	0.20
			16.0	16.0	512	1.49	0.16
			32.0	32.0	1024	1.49	0.20
			1.0	-1.0	16.0	16.0	512
	2.0	-2.0	16.0	16.0	1024	1.50	0.17
	1.0	-1.0	32.0	32.0	256	1.53	0.16
	2.0	-2.0	16.0	16.0	256	1.54	0.17
	1.0	-1.0	16.0	16.0	256	1.54	0.18
250	1.0	-1.0	32.0	32.0	1024	2.73	0.02
			32.0	32.0	512	2.87	0.01
			16.0	16.0	1024	2.90	0.02
			1.0	-1.0	16.0	16.0	1024
	2.0	-2.0	32.0	32.0	1024	2.93	0.03
			16.0	16.0	512	3.07	0.02
			32.0	32.0	256	3.08	0.01
			1.0	-1.0	32.0	32.0	512
			16.0	16.0	512	3.15	0.02
			32.0	32.0	256	3.15	0.01

Table S1: **Higher-order oscillator simulation result for top parameter configurations.** The mean energy and fraction of satisfied instances are listed for the top 10 parameter configurations for each problem size obtained from simulations of higher-order oscillator networks.

# of variables	ΔE_{\min}	λ	ρ	r	q_{\max}	# of cycles	energy	satisfied probability	
20	5	1.0	-1.0	0.10	0.10	1024	0.66	0.51	
	10	1.0	-1.0	0.10	0.10	1024	0.66	0.52	
	5	1.0	-1.0	0.10	0.10	512	0.67	0.52	
	10	1.0	-1.0	0.10	0.10	512	0.71	0.50	
	1	1.0	-1.0	1.10	1.10	128	0.77	0.44	
					1.05	1.05	128	0.77	0.44
	10	0.5	-0.5	0.10	0.10	1024	0.77	0.47	
		2.0	-2.0	0.10	0.10	1024	0.78	0.46	
					1.00	1.00	64	0.80	0.40
							128	0.80	0.40
50	10	2.0	-2.0	1.00	1.00	128	1.30	0.20	
		0.5	-0.5	0.10	0.10	512	1.37	0.21	
		1.0	-1.0	1.00	1.00	64	1.38	0.18	
		2.0	-2.0	1.00	1.00	64	1.42	0.20	
					0.10	0.10	512	1.44	0.22
		1.0	-1.0	1.00	1.00	128	1.48	0.18	
		2.0	-2.0	0.10	0.10	256	1.60	0.15	
		1.0	-1.0	0.10	0.10	256	1.60	0.15	
	5	1.0	-1.0	1.00	1.00	1024	1.62	0.17	
						512	1.66	0.16	
100	10	2.0	-2.0	1.00	1.00	128	2.82	0.08	
		1.0	-1.0	1.00	1.00	128	3.02	0.05	
		2.0	-2.0	1.00	1.00	64	3.02	0.05	
		1.0	-1.0	1.00	1.00	64	3.20	0.05	
		0.5	-0.5	0.10	0.10	512	3.54	0.02	
	5	1.0	-1.0	1.00	1.00	1024	3.59	0.04	
						512	3.62	0.04	
						256	3.66	0.04	
						128	3.69	0.03	
			2.0	-2.0	1.00	1.00	64	3.76	0.02
250	10	2.0	-2.0	1.00	1.00	32	9.00	0.00	
	5	1.0	-1.0	1.00	1.00	128	9.31	0.00	
						512	9.85	0.00	
						1024	9.94	0.00	
	10	2.0	-2.0	0.10	0.10	1024	10.00	0.00	
		0.5	-0.5	0.10	0.10	512	10.02	0.00	
	5	1.0	-1.0	1.00	1.00	256	10.27	0.00	
	10	2.0	-2.0	1.00	1.00	64	10.50	0.00	
		1.0	-1.0	1.00	1.00	64	10.60	0.00	
	5	2.0	-2.0	1.00	1.00	64	10.70	0.00	

Table S2: **Second-order oscillator simulation result for top parameter configurations.** The mean energy and fraction of satisfied instances are listed for the top 10 parameter configurations for each problem size obtained from simulations of second-order oscillator networks.

# of variables	r	q_{\max}	# of cycles	energy	satisfied probability	
20	1	0.000	256	1.43	0.25	
		0.001	256	1.43	0.25	
	10	0.000	16	1.43	0.25	
			64	1.43	0.25	
		0.001	64	1.43	0.25	
			64	1.43	0.25	
	20	0.010	64	1.43	0.25	
			64	1.43	0.25	
		0.000	64	1.43	0.25	
			64	1.43	0.25	
0.001		64	1.43	0.25		
		64	1.43	0.25		
50	20	0.010	64	3.45	0.00	
		64	3.45	0.00		
	50	0.010	64	3.45	0.00	
		64	3.45	0.00		
	100	0.001	256	3.45	0.00	
			1024	3.45	0.00	
		0.010	64	3.45	0.00	
			256	3.45	0.00	
		0.010	1024	3.45	0.00	
			64	3.45	0.00	
		0.100	64	3.45	0.00	
			1024	3.45	0.00	
	100	10	0.000	64	3.46	0.00
			64	3.46	0.00	
100		0.001	256	6.81	0.00	
			1024	6.81	0.00	
		0.000	1024	6.82	0.00	
			256	6.83	0.00	
		0.001	64	6.83	0.00	
			256	6.83	0.00	
		0.010	256	6.83	0.00	
			1024	6.83	0.00	
50		0.000	64	6.84	0.00	
		64	6.84	0.00		
250		50	0.001	64	6.85	0.00
			64	14.31	0.00	
	100	0.010	64	15.82	0.00	
			1024	15.82	0.00	
		0.000	256	15.84	0.00	
			1024	15.84	0.00	
		0.010	1024	15.84	0.00	
			64	15.85	0.00	
		0.000	1024	15.85	0.00	
			64	15.86	0.00	
	0.010	256	15.86	0.00		
		1024	15.86	0.00		
	0.100	1024	15.86	0.00		
		256	15.86	0.00		
0.000	256	15.87	0.00			
	64	15.90	0.00			

Table S3: **Higher-order Kuramoto oscillator simulation result for top parameter configurations.** The mean energy and fraction of satisfied instances are listed for the top 10 parameter configurations for each problem size obtained from simulations of higher-order Kuramoto oscillator networks.

REVIEWERS' COMMENTS

Reviewer #3 (Remarks to the Author):

The authors addressed my previous concerns.

Response to the second round of comments on “Efficient Optimization with Higher-Order Ising Machines”

Reviewer 3

The authors addressed my previous concerns

We thank the reviewer for their help to improve the manuscript.